# Uniformly Distributed Feature Representations for Fair and Robust Learning

**Kiran Krishnamachari**  *kirank@u.nus.edu*
*Institute for Infocomm Research ($I^2R$), A\*STAR, Singapore*
*School of Computing, National University of Singapore, Singapore*

**See-Kiong Ng**  *seekiong@nus.edu.sg*
*Institute of Data Science, National University of Singapore, Singapore*
*School of Computing, National University of Singapore, Singapore*

**Chuan-Sheng Foo**  *foo__chuan__sheng@i2r.a-star.edu.sg*
*Institute for Infocomm Research ($I^2R$), A\*STAR, Singapore*
*Centre for Frontier AI Research (CFAR), A\*STAR, Singapore*

**Reviewed on OpenReview:** *https://openreview.net/forum?id=PgLbS5yp8n*

## Abstract

A fundamental challenge in machine learning is training models that generalize well to distributions different from the training distribution. Empirical Risk Minimization (ERM), which is the predominant learning principle, is known to under-perform in minority sub-populations and fail to generalize well in unseen test domains. In this work, we propose a novel learning principle called Uniform Risk Minimization (URM) to alleviate these issues. We first show theoretically that uniform training data distributions and feature representations support robustness to distribution shifts. Motivated by this result, we propose an empirical method that trains deep neural networks to learn a uniformly distributed feature representation in their final activation layer for improved robustness. Our experiments on multiple datasets for sub-population shifts and domain generalization show that URM improves the generalization of deep neural networks without requiring knowledge of groups or domains during training. URM is competitive with the best existing methods designed for these tasks and can also be easily combined with them for improved performance. Our work sheds light on the importance of the distribution of learned feature representations for model robustness and fairness. Code is available at https://github.com/kiranchari/UniformRiskMinimization.

## 1 Introduction

ERM has been the predominant approach to train machine learning models. Consistently achieving high accuracy on test distributions identical to the training data distribution was a challenge until the resurgent success of deep learning. However, machine learning practitioners today realize that a major challenge is robustness in test distributions different from the training data distribution. Hence, the assumption that training and test data distributions are identical no longer captures the challenge of modern machine learning.

In this work, we consider the challenging problem of training models that perform well in distributions different from the training distribution without assuming knowledge of the test distribution during the learning process. Specifically, we explore the question of what is the best data distribution to train classifiers on for improved downstream generalization. We consider two downstream generalization tasks in our work, i.e., sub-population shifts and domain generalization. In the sub-population or group shift problem setup, we have input $\mathbf{x} \in \mathcal{X}$ and labels $y \in \mathcal{Y}$, with the objective to learn a function $f : \mathcal{X} \rightarrow \mathcal{Y}$. Moreover, there exist attributes of the data $a_1, ..., a_i, ..., a_m$, $a_i \in \mathcal{A}_i$, which may not be available during training.

Hence, distinct sub-populations or groups of the data can be defined using combinations of the attribute and label, for example $h : \mathcal{A} \times \mathcal{Y} \to \mathcal{G}$. The training distribution over $(\mathbf{x}, y)$ is a mixture of group-specific distributions $p_g$ i.e. $p_{tr} = \sum_{g \in \mathcal{G}} \gamma_g p_g$, where $\gamma \in \Delta_{|\mathcal{G}|}$. The test distribution, which is not observed during training, is: $p_{te} = \sum_{g \in \mathcal{G}} \beta_g p_g$, where $\beta \in \Delta_{|\mathcal{G}|}$ and $\gamma \neq \beta$. The objective for $f$ is to perform accurately in the unobserved $p_{te}$. Generally, models perform most poorly when the test distribution only comprises minority sub-populations of the training distribution (Hashimoto et al., 2018; Zhang et al., 2020). In sensitive applications of machine learning such as healthcare, this leads to *fairness* issues when models perform better for some groups of people than others. Multiple methods have been proposed to tackle this problem of robustness to sub-population shifts. A common approach so far has been identifying the samples where models performs poorly and up-weighting these samples during training. This *identify-and-emphasise* paradigm has been proposed in multiple works (Liu et al., 2021; Zhang & Sabuncu, 2018; Yaghoobzadeh et al., 2021; Sanh et al., 2021; Utama et al., 2020). Many proposed methods also rely on the knowledge of minority sub-population attributes for training or validation to achieve good performance (Gowda et al., 2021; Izmailov et al., 2022; Menon et al., 2021; Nam et al., 2022; Yao et al., 2022). However, knowledge of relevant sub-populations may not always be known, especially in novel settings, so this assumption may be impractical.

In domain generalization (DG), models are provided a set of training domains, each containing examples about the same task, with the goal that they must perform well on unseen target domains (Blanchard et al., 2011; Muandet et al., 2013). While challenging, DG captures real-world scenarios where unforeseen distribution shifts between training and test distributions can be encountered by models. Both scenarios, sub-population shifts and domain generalization, require novel solutions that enable generalization while requiring limited privileged information such as access to minority group annotations or access to samples from the test domains.

In this work, we present a novel learning principle called Uniform Risk Minimization (URM) to alleviate these issues. We first propose a novel risk measure, *uniform risk*, which is defined as the *expected* risk of the model as the test distribution varies over the support of the training distribution. Specifically, uniform risk is equal to the expected risk of a model over all possible test distributions under a uniform prior. Our main theoretical result is as follows: *training classifiers on uniformly distributed training data or learned feature representations is an optimal choice to lower uniform risk and supports robustness and fairness.* Motivated by this result, we propose an empirical method to encourage deep neural networks to learn uniformly distributed feature representations using an adversarial training objective. We show using experiments that encouraging models to have uniformly distributed feature representations significantly improves their robustness to sub-population shifts and domain generalization, without requiring knowledge of groups or domains.

The major contributions of our work are as follows:

1. We propose **Uniform Risk Minimization (URM)**, a novel learning principle for out of distribution (ood) robustness and fairness (Section 2.3).

2. We first theoretically show that training classifiers on **uniformly distributed input data or feature representations in deep neural networks** is optimal for lowering *uniform risk* and supports robustness (Propositions 2.2, 2.3).

3. Motivated by our theoretical analysis, we propose a **method to learn uniformly distributed feature representations** in the final activation layer of deep neural networks using an adversarial objective (Section 2.6). We empirically demonstrate the efficacy of this method for sub-population shifts and domain generalization, without requiring knowledge of task-specific knowledge such as group or domain labels (Section 3).

## 2 Uniform Risk Minimization

In this section, we present a novel learning principle called Uniform Risk Minimization (URM) for robustness under distribution shifts. Motivated by real-world scenarios where we do not necessarily have knowledge of the environments models will be tested on, we propose a novel risk measure called *uniform risk* that is the

*expected* test risk over *all* possible test distributions under a uniform prior (Section 2.3). We then show theoretically that uniformly distributed training data (Proposition 2.2) or feature representations (Proposition 2.3) are optimal in lowering uniform risk. We then present a practical method for deep learning models that encourages their feature representations to be uniformly distributed using adversarial distribution matching to improve their robustness in various distribution shift scenarios (Section 2.6).

## 2.1 Notation and Problem Setup

We describe the general setting of distribution shifts between the training and test distributions for the classification problem. We denote the joint training distribution as $p_{tr}(x, y)$ and the test distribution as $p_{te}(x, y)$, where $x$ denotes the input and $y$ denotes the label. We assume these two distributions have the same support sets $\mathcal{X}, \mathcal{Y}$. We denote a labeled training dataset of size $N_{tr}$ sampled from the training distribution $(x_{tr}^{(i)}, y_{tr}^{(i)})_{i=1}^{N_{tr}}$, where $(x_{tr}^{(i)}, y_{tr}^{(i)}) \sim p_{tr}(x, y)$, and a dataset of size $N_{te}$ from the test distribution $(x_{te}^{(i)}, y_{te}^{(i)})_{i=1}^{N_{te}}$, where $(x_{te}^{(i)}, y_{te}^{(i)}) \sim p_{te}(x)$.

## 2.2 A Generalization Bound for Distribution Shifts

We first define the test loss under distribution shift. The negative log-loss on the test distribution is:

$$l_{test} = \mathbb{E}_{p_{te}(x,y)}[-\log \hat{p}(y|x)] \tag{1}$$

Next, we propose a bound of the test loss using the training loss ($l_{train}$) and the Kullback-Leibler (KL) divergence between train and test distributions by extending Proposition 1 in Nguyen et al. (2022).

**Proposition 2.1.** *If the loss* $-\log \hat{p}(y|x)$ *is bounded by* $M$[1] *and the labeling mechanism,* $p(y|x)$*, is unchanged between train and test distributions (covariate-shift assumption), we have:*

$$l_{test} \leq l_{train} + \frac{M}{\sqrt{2}}[\text{KL}[p_{te}(x)|p_{tr}(x)] + \frac{M}{4\sqrt{2}} \tag{2}$$

*Proof.* Proof in Appendix A. □

An alternate proposition based on the *label* or *target*-shift assumption is discussed in Appendix E.1.

## 2.3 Uniform Risk

In real-world deployments of machine learning, we do not always know which test distributions a model will encounter. Hence, we propose a novel risk measure that incorporates the *expected* risk over all possible test distributions. As we assume that we do not know the test distribution, we wish to train models that perform well under various scenarios. So the expected risk is defined using an uninformative *uniform prior* over all possible test distributions. A key fact enabling the mathematical definition of such a uniform prior is that all computer systems represent data using finite-precision floating points, making all data representations discrete in practice. This enables us to define the uniform prior over all test distributions using the Dirichlet distribution, which is the natural prior distribution for any discrete distribution. Let $N$ be the size of the support set of the data distribution, i.e. the total number of possible inputs. For $N$-dimensional distributions, the uniform prior can be represented using the Dirichlet distribution such that $p_{te}(x) \sim Dir(\boldsymbol{\alpha} = \mathbb{1})$. Note that $Dir(\boldsymbol{\alpha} = \mathbb{1})$ is the uniform prior with parameters $\alpha_i = 1, \forall i \in \{1, \ldots, N\}$, and $\alpha_0 = \sum_{i=1}^{N} \alpha_i = N$. In Eq. 3, we define *uniform risk*, $\mathcal{R}_U$, as the expectation of the test loss under a uniform prior over all possible test distributions. We assume that training and test distributions have the same support.

---

[1]As suggested by (Nguyen et al., 2022), in a classification problem, this can be enforced easily by augmenting the output softmax of the classifier such that each class probability is always at least $\exp(-M)$. For example, if we choose $M = 3 \Rightarrow \exp(-M) \approx 0.05$, and if the output softmax is $(p_1, p_2, ..., p_C)$, we can augment it as $(p_1 \cdot K + 0.05, p_2 \cdot K + 0.05, ..., p_C \cdot K + 0.05)$, where $K = 1 - 0.05 \cdot C$ and $C$ is the number of classes. This ensures the bound for the loss on a sample, while leaving the output prediction class unchanged.

$$\mathcal{R}_U := \mathbb{E}_{p_{te}(x) \sim Dir(\boldsymbol{\alpha}=\mathbb{1})}[l_{test}] \tag{3}$$

We now upper bound *uniform risk* using Eq. 2:

$$\mathbb{E}_{p_{te}(x) \sim Dir(\boldsymbol{\alpha}=\mathbb{1})}[l_{test}] \leq l_{train} + \frac{M}{\sqrt{2}}\mathbb{E}_{p_{te}(x) \sim Dir(\boldsymbol{\alpha}=\mathbb{1})}[\mathrm{KL}[p_{te}(x)|p_{tr}(x)]] + \frac{M}{4\sqrt{2}} \tag{4}$$

Note that the bound depends on the training loss, the expected KL divergence between training and test distributions, and a constant (Equation 4). Modern deep learning systems can minimize the training loss nearly to zero as long as the task is well defined and realizable by modern architectures. Hence, the main component of Eq. 4 to minimize would be the *expected KL divergence between the training and test distributions*. This leads us to our first key result:

**Proposition 2.2.** *The expected KL divergence between the train and test distributions in Eq. 4 is minimized when the training data distribution $p_{tr}(x)$ is uniform:*

$$\underset{p_{tr}(x)}{\arg\min}\, \mathbb{E}_{p_{te}(x) \sim Dir(\boldsymbol{\alpha}=\mathbb{1})}[\mathrm{KL}[p_{te}(x)|p_{tr}(x)]] = u_{tr}^*(x)$$

$$where\; u_{tr}^*(x_i) = \frac{1}{N}, \forall i \in \{1, \ldots, N\}$$

*Proof.* Proof in Appendix B. □

Proposition 2.2 states that the training data distribution that minimises the upper bound on *uniform risk* is the uniform distribution, $u_{tr}^*(x)$. This means that training on uniform data distributions may improve robustness to distribution shifts. However, most practical training datasets may not be uniformly distributed. While uniformity could in principle be achieved by re-weighting the data distribution i.e. by down-weighting high-density regions and up-weighting low-density regions of the distribution, this would require knowledge of the data distribution or an accurate estimation of it, which is non-trivial in practice. Modern deep learning systems are usually trained on high-dimensional image or audio datasets whose distributions are not known and are difficult to estimate. Hence, we will consider a similar proposition using the learned feature representations of deep neural networks (Section 2.5).

## 2.4 Revisiting Group-balanced and Class-balanced Training via URM

Rather than considering the data distribution over the input space, we can also consider *sub-group* and *class* distribution shifts i.e., shifts in $p(g)$ and $p(y)$, respectively. Similar to Proposition 2.2, we can show that training with uniform sub-group or class distributions—equivalent to group-balanced and class-balanced training—reduces uniform risk as defined in these scenarios (discussion and proof in Appendix E). Additionally, in these scenarios, *uniform risk* can be shown to be equivalent to the balanced risk across sub-groups or classes, respectively (proof in Appendix E). Thus, minimizing uniform risk in these scenarios directly corresponds to minimizing the balanced risk across sub-groups or classes. In fact, prior work (Idrissi et al., 2022) has shown that group-balanced training is an effective approach for robustness to sub-population shifts when group-attributes of the training samples are known. Prior work (Buda et al., 2018) has also demonstrated the efficacy of class-balanced training. We provide further empirical evidence below that class-balanced training is effective for label-shifts in Section 3.3. Hence, URM provides a unified perspective on various algorithms that perform group or class balanced training.

## 2.5 Lowering Uniform Risk in Learned Feature Representation Spaces

We have above defined uniform risk using the input space (Eq. 3) and shown that uniform training data distributions lower uniform risk (Proposition 2.2). To implement this in practice would require training our

models on uniformly distributed training data distributions. However, most practical training datasets are not uniformly distributed and it is difficult to modify their distribution to match a uniform distribution. Thus, we propose a re-definition of the bound on *uniform risk* using the *learned* feature representation space of a deep model. Hence, the focus shifts from identifying the optimal input training data distribution to determining the optimal latent variable distribution. Note that the latent variable is not fixed as it is learned by a model and hence, can be regularized during training to attain a specific distribution. As in Proposition 2.2, we can show that uniformly distributed *feature representations* also lower uniform risk. This allows us to implement URM by training deep neural networks to *learn* a uniformly distributed feature representations rather than training on an input data distribution that is already uniformly distributed.

Let $G$ be a model parameterized by a deep neural network that outputs a latent variable $z$ i.e. $z = G(x)$, which has dimensionality $Z$ and is used by a downstream task head such as linear classifier. In order to extend the propositions to the latent space, $z$, we make additional assumptions about the input data $x, y$ and the latent variable, $z$, following Nguyen et al. (2022). The key assumption is that $I_{tr}(z, y) = I_{tr}(x, y)$ where $I(\cdot, \cdot)$ is mutual information. This is a reasonable assumption that the latent variable $z$ is *sufficient* to solve the task and contains the same information about the output variable $y$ as does the input variable $x$. As the latent representation $z$ is learned while optimizing the downstream task objective, it must learn to retain information about the label. The remaining assumptions are weak assumptions about the data distribution and the standard covariate-shift assumption (all assumptions are detailed in Appendix D). Using these assumptions, Proposition 1 of Nguyen et al. (2022) can be re-written as follows:

$$l_{test} \leq l_{train} + \frac{M}{\sqrt{2}}[\text{KL}[p_{te}(z)|p_{tr}(z)] + \frac{M}{4\sqrt{2}} \tag{5}$$

We have thus replaced $x$ (input space) with $z$ (latent space). The rest of our claims in Propositions 2.1 and 2.2 can then straightforwardly applied to the latent space rather than the input space. Uniform risk can be bounded using the latent feature representation space as follows,

$$\mathbb{E}_{p_{te} \sim Dir(\boldsymbol{\alpha}=\mathbb{1})}[l_{test}] \leq l_{train} + \frac{M}{\sqrt{2}}\mathbb{E}_{p_{te}(z) \sim Dir(\boldsymbol{\alpha}=\mathbb{1})}[\text{KL}[p_{te}(z)|p_{tr}(z)]] + \frac{M}{4\sqrt{2}} \tag{6}$$

As above, we aim to minimize the expected divergence between the training and test feature distributions in Eq. 6. This leads us to our second key result: *uniform feature distributions minimize the expected divergence between training and test feature distributions* (Proposition 2.3).

**Proposition 2.3.** *The expected KL-divergence between train and test distributions in Eq. 6 is minimized when the training feature distribution $p_{tr}(z)$ is uniform:*

$$\underset{p_{tr}(z)}{\arg\min} \, \mathbb{E}_{p_{te}(z) \sim Dir(\boldsymbol{\alpha}=\mathbb{1})}[\text{KL}[p_{te}(z)|p_{tr}(z)]] = u_{tr}^*(z)$$

$$where \; u_{tr}^*(z_i) = \frac{1}{Z}, \forall i \in \{1, \dots, Z\}$$

*Proof.* Follows proof of Proposition 2.2. □

Hence, we have shown that uniformly distributed *feature* representations also lower *uniform risk*. This motivates our proposed method below that encourages deep neural networks to learn a feature representation space that is uniformly distributed for improved robustness (Section 2.6). Note that the assumption $I_{tr}(z, y) = I_{tr}(x, y)$ ensures that the representation $z$ is useful for classification while being uniformly distributed. In practice, this is ensured by the joint optimization of the classifier and our proposed regularizer objectives.

## 2.6 Encouraging Feature Representation Uniformity using Adversarial Distribution Matching

Motivated by Proposition 2.3, we perform URM by learning deep models with uniformly distributed feature representations. In order to encourage the distribution of deep feature vectors to match a uniform distribution, we use an adversarial training approach. Adversarial training is a useful approach to match distributions as it does not require significant modifications to the output of the encoder. Alternate approaches such as minimizing KL-divergence would require limiting the expressivity of the model to output a parametric distribution in order to analytically compute its divergence from a target distribution. We train a *domain classifier* (or *discriminator*) to distinguish between feature vectors produced by an encoder and samples from a uniform noise distribution. Simultaneously, the encoder is trained to fool the domain classifier by generating feature vectors that resemble a uniform distribution. This is achieved through adversarial training: the domain discriminator, $D$, learns to dif-

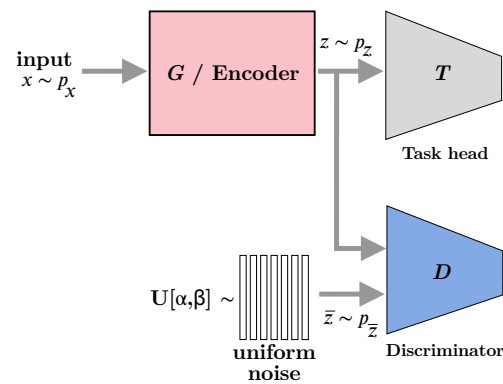

Figure 1: Overview of adversarial distribution matching to encourage the encoder to output uniformly distributed representations.

ferentiate encoded feature vectors from random uniform noise, while the encoder, $G$, is optimized to both confuse $D$ and learn useful representations for a downstream task, typically handled by a linear classifier $T$ (see Figure 1 for an overview). Our method resembles Adversarial Autoencoders (Makhzani et al., 2016), which use adversarial training to align the latent space of autoencoders with desired prior distributions.

During training, we want the discriminator $D$'s predictions for uniform random noise vectors to be accurate by maximizing $\mathbb{E}_{\bar{z} \sim p_u}[\log D(\bar{z})]$. On the other hand, given an encoded feature vector $G(x)$ the discriminator must output a probability, $D(G(x))$, close to zero by maximizing $\mathbb{E}_{x \sim p_x}[\log(1 - D(G(x)))]$. The encoder must be trained to fool $D$ to produce a high probability for feature vectors by minimizing $\mathbb{E}_{x \sim p_x}[\log(1 - D(G(x)))]$ or maximising $\mathbb{E}_{x \sim p_x}[\log D(G(x))]$. The combined objective function to update $D$ and $G$ represents a *minimax game* with the following loss function:

$$\min_G \max_D L(D, G) = \mathbb{E}_{\bar{z} \sim p_u}[\log D(\bar{z})] + \mathbb{E}_{z \sim p_z}[\log(1 - D(z)]$$

where $\mathbb{E}_{\bar{z} \sim p_u}[\log D(\bar{z})]$ is not used to update $G$. Generative adversarial networks (GAN) (Goodfellow et al., 2014) theory shows that the above objective minimizes the Jensen–Shannon Divergence between the encoded feature distribution and the uniform distribution when the discriminator is optimal. Let the $\theta$ be the parameters of the task model, which includes the encoder G as its backbone and a task head $T$ such as a linear classifier i.e. $\theta = \{G, T\}$. $\mathcal{L}_T$ is the task-specific loss function. We can summarize the overall URM training objective for $\theta$ and D as follows:

$$\min_\theta \max_D \mathcal{L}(\theta, D) = \mathbb{E}_{(x,y) \sim p_{tr}}[\mathcal{L}_T(T(G(x)), y)] + \lambda \mathbb{E}_{x \sim p_{tr}}[\log(1 - D(G(x))] + \mathbb{E}_{\bar{z} \sim p_u}[\log D(\bar{z})]$$

The hyper-parameter $\lambda$ determines the weight for training the encoder and discriminator to classify the feature outputs $z = G(x)$. A higher $\lambda$ increases the strength of regularization at the expense of feature learning for the downstream task and lower $\lambda$ weakens regularization but allows the encoder to pay more attention to the downstream task. The same $\lambda$ is applied to both update the discriminator and encoder so that one does not overpower the other. The choice of activation function applied to the output of the encoder G can be changed for each task. We choose from either the hyperbolic Tangent (TanH) or ReLU activations. In case of TanH activations, the uniform noise distribution is $\bar{z} \sim \mathcal{U}(-1, 1)$ and in case of ReLU it is $\bar{z} \sim \mathcal{U}(0, 1)$. We use Leaky ReLUs in the discriminator to improve gradient flow to the generator. We train the encoder G and discriminator D alternately.

# 3 Experiments

## 3.1 Experimental Settings and Datasets

For group robustness, our experimental setup followed Yang et al. (2023). Specifically, we used the evaluation setting where group attributes are unknown in the training set. The *worst-class* accuracy is used to perform model selection, i.e., choice of the best checkpoint from each training run, as it was found to be an effective alternative in the absence of group-attributes by Yang et al. (2023). For all methods, group attributes in the validation set were used to select hyper-parameters as well as to fine-tune methods that require a group-balanced dataset e.g. DFR (Izmailov et al., 2022). We report *worst-group* and *balanced* (across classes) accuracies. All results were averaged across three random seeds. For domain generalization experiments, our experimental setup followed Gulrajani & Lopez-Paz (2021) using the *leave-one-domain-out* cross-validation scheme. We report the average accuracy across all test domains. For both tasks, we searched over sixteen random hyper-parameter combinations and applied adversarial distribution matching to the penultimate layer before the linear classification head. Hyper-parameter search ranges for URM are included in the Appendix (Table 4). Adversarial training is often unstable in certain contexts, such as generative modeling. However, in our experiments, we did not encounter significant instability. This is likely because we are not generating high-dimensional images, as in GANs, which involve a more complex distribution-matching task. Instead, we are generating lower-dimensional feature vectors to align with a uniform distribution. For group robustness, we evaluate on Waterbirds (Wah et al., 2011), CelebA (Liu et al., 2015), CivilComments (Borkan et al., 2019) and MultiNLI (Williams et al., 2018). For Waterbirds and CelebA, we use the ResNet50 (He et al., 2016) architecture pretrained on ImageNet1k (Russakovsky et al., 2015). For MultiNLI and CivilComments, we use BERT (Devlin et al., 2019) pretrained on Book Corpus and English Wikipedia data. For domain generalization (DG), we benchmark on ColoredMNIST (Arjovsky et al., 2020), RotatedMNIST (Ghifary et al., 2015), PACS (Li et al., 2017), VLCS (Fang et al., 2013), OfficeHome (Venkateswara et al., 2017) and TerraIncognita (Beery et al., 2018). For DG benchmarks, all methods used the ResNet50 (He et al., 2016) architecture pretrained on ImageNet1k (Russakovsky et al., 2015). Other training and dataset parameters are based on (Gulrajani & Lopez-Paz, 2021). We further describe all benchmark datasets in Appendix F. All experiments were run on a computer with 8 A100 GPUs.

## 3.2 Baselines

We consider baselines including ERM and various methods proposed for sub-population shifts and domain generalization. For sub-population shifts, we consider GroupDRO, LfF, JTT, LISA, DFR, Mixup, IRM, CORAL, MMD, DANN, C-DANN, ReSample, ReWeight, Focal Loss, CBLoss (Class-balanced loss), LDAM, Balanced Softmax (BSoftmax), CRT and ReWeightCRT. We follow Yang et al. (2023) for implementation of these baselines. For domain generalization, we consider Inter-domain Mixup (Mixup), MLDF, MTL, ARM, SagNet, RSC, and VREx. We follow Gulrajani & Lopez-Paz (2021) for implementation of these baselines. For a description of these baselines, please refer to our Appendix G. We note that URM is a generic robust representation learning method that does not require any knowledge of training domains or groups for both sub-population shifts and domain generalization.

## 3.3 URM Supports Class-balanced Training for Robustness under Label-shifts

As discussed in Section 2.4, when considering shifts in the class-distribution (*label-shifts*) alone, we demonstrate that URM, which corresponds to class-balancing, produces the best *balanced accuracy*. We consider the binary classification tasks in the Waterbirds and ColoredMNIST datasets. We trained models with different class-distributions and compute each model's *balanced accuracy*, which is the average of the class-specific accuracies on the test set. Balanced accuracy is a commonly used metric in imbalanced classification tasks. We train models with different values of $p$ ($p \in \{0.1, 0.2, 0.3, 0.4, 0.5, 0.6, 0.7, 0.8, 0.9\}$), where $p$ corresponds to the binary class distribution parameter (Table 1). Note that $p = 0.5$ corresponds to class-balanced training, which corresponds to URM for label shift scenarios. The URM model ($p = 0.5$) achieves the highest *balanced accuracy* on the test set (Table 1). *Balanced accuracy* deteriorates when the the training class distribution is highly imbalanced e.g. $p = 0.9$ or $0.1$. As discussed in Section 2.4, *uniform risk* corresponds to

balanced accuracy across classes in the label-shift scenario, hence URM (class-balancing) is a useful strategy in dealing with unexpected label shifts. Our results further support and provide a unified perspective on the common-practice of class-balanced training.

### 3.4 Uniformly Distributed Feature Representations Improve Sub-group Robustness

URM learns more sub-group robust representations compared to ERM on multiple datasets (Table 2). URM can also be easily combined with other methods as it is a generic representation learning method. For example, we combined URM with Deep Feature Reweighting (DFR) as it involves fine-tuning the classifier on a group balanced validation set, which URM does not require. Notably, when combined with DFR, URM achieved the best worst-group accuracy on the Waterbirds and MultiNLI datasets (URM (DFR), Table 2). On MultiNLI, URM (DFR) achieved 4% higher worst-group accuracy than the existing best methods. URM also achieved the highest balanced accuracy on MultiNLI. The better worst-group accuracy achieved by URM

Table 1: *Balanced accuracies (%)* after training with different class distributions. Results are averaged across three random seeds.

| TRAINING | WATERBIRDS | COLOREDMNIST |
|---|---|---|
| $p = 0.1$ | 85.7 | 55.3 |
| $p = 0.2$ | 86.0 | 68.0 |
| $p = 0.3$ | 85.3 | 76.5 |
| $p = 0.4$ | 86.1 | **76.9** |
| $p = 0.5$ (URM) | **86.8** | **76.9** |
| $p = 0.6$ | 85.7 | 76.5 |
| $p = 0.7$ | 86.2 | 76.3 |
| $p = 0.8$ | 86.4 | 75.7 |
| $p = 0.9$ | 85.7 | 66.6 |

(DFR) reflects the improved representation learned by URM in the absence of any group attribute information. We note that URM is a generic robust representation learning method and is not specific to sub-group robustness like many of the other baseline methods.

Table 2: Worst-group and balanced accuracies on group robustness benchmarks. Results for other methods are from Yang et al. (2023).

| Algorithm | Waterbirds | | CelebA | | CivilComments | | MultiNLI | |
|---|---|---|---|---|---|---|---|---|
| | worst (%) | balanced (%) | worst (%) | balanced (%) | worst (%) | balanced (%) | worst (%) | balanced (%) |
| ERM | 69.1 ±4.7 | 83.1 ±2.0 | 57.6 ±0.8 | 93.0 ±0.0 | 63.2 ±1.2 | 79.4 ±0.2 | 66.4 ±2.3 | 81.0 ±0.3 |
| Mixup | 77.5 ±0.7 | 88.9 ±0.2 | 57.8 ±0.8 | 93.1 ±0.1 | 65.8 ±1.5 | 79.7 ±0.0 | 66.8 ±0.3 | 81.7 ±0.1 |
| GroupDRO | 73.1 ±0.4 | 86.3 ±0.5 | 68.3 ±0.9 | 93.9 ±0.1 | 61.5 ±1.8 | 81.3 ±0.1 | 64.1 ±0.8 | 81.1 ±0.2 |
| CVaRDRO | 75.5 ±2.2 | 88.5 ±0.3 | 60.2 ±3.0 | 93.1 ±0.1 | 62.9 ±3.8 | 80.8 ±0.1 | 48.2 ±3.4 | 75.4 ±0.2 |
| JTT | 71.2 ±0.5 | 86.8 ±0.2 | 48.3 ±1.5 | 92.4 ±0.2 | 51.0 ±4.2 | 77.7 ±0.8 | 65.1 ±1.6 | 81.4 ±0.0 |
| LfF | 75.0 ±0.7 | 86.3 ±0.3 | 53.0 ±4.3 | 85.3 ±2.9 | 42.2 ±7.2 | 69.7 ±4.7 | 57.3 ±5.7 | 71.4 ±1.6 |
| LISA | 77.5 ±0.7 | 88.9 ±0.2 | 57.8 ±0.8 | 93.1 ±0.1 | 65.8 ±1.5 | 79.7 ±0.0 | 66.8 ±0.3 | 81.7 ±0.1 |
| ReSample | 70.0 ±1.0 | 85.0 ±0.2 | **74.1** ±**2.2** | 93.8 ±0.1 | 61.0 ±0.6 | 80.7 ±0.1 | 66.8 ±0.5 | 81.5 ±0.0 |
| ReWeight | 71.9 ±0.6 | 86.2 ±0.1 | 69.6 ±0.2 | 94.0 ±0.1 | 59.3 ±1.1 | 81.3 ±0.0 | 64.2 ±1.9 | 79.4 ±0.2 |
| SqrtReWeight | 71.0 ±1.4 | 87.2 ±0.6 | 66.9 ±2.2 | 93.9 ±0.1 | **68.6** ±**1.1** | 80.6 ±0.2 | 63.8 ±2.4 | 80.6 ±0.2 |
| CBLoss | 74.4 ±1.2 | 86.2 ±0.6 | 65.4 ±1.4 | 93.8 ±0.0 | 67.3 ±0.2 | 80.3 ±0.3 | 63.6 ±2.4 | 80.6 ±0.3 |
| Focal | 71.6 ±0.8 | 87.1 ±0.3 | 56.9 ±3.4 | 92.6 ±0.3 | 61.9 ±1.1 | 78.7 ±0.3 | 62.4 ±2.0 | 80.9 ±0.2 |
| LDAM | 70.9 ±1.7 | 86.0 ±0.2 | 57.0 ±4.1 | 93.2 ±0.2 | 28.4 ±7.7 | 69.5 ±3.2 | 65.5 ±0.8 | 80.9 ±0.1 |
| BSoftmax | 74.1 ±0.9 | 87.0 ±1.0 | 69.6 ±1.2 | **94.2** ±**0.1** | 58.3 ±1.1 | 81.1 ±0.1 | 63.6 ±2.4 | 80.6 ±0.2 |
| DFR | 89.0 ±0.2 | 91.2 ±0.1 | 73.7 ±0.8 | 93.2 ±0.0 | 64.4 ±0.1 | 79.0 ±0.0 | 63.8 ±0.0 | 80.2 ±0.0 |
| CRT | 76.3 ±0.8 | 87.9 ±0.1 | 69.6 ±0.7 | 93.6 ±0.0 | 67.8 ±0.3 | 80.7 ±0.0 | 65.4 ±0.2 | 80.2 ±0.0 |
| ReWeightCRT | 76.3 ±0.2 | 88.0 ±0.2 | 70.7 ±0.6 | 93.6 ±0.0 | 64.7 ±0.2 | 80.7 ±0.0 | 65.2 ±0.2 | 80.2 ±0.0 |
| URM | 76.9 ±1.9 | 87.3 ±0.7 | 65.6 ±1.2 | 93.4 ±0.3 | 66.2 ±0.5 | 80.1 ±0.4 | 67.7 ±0.7 | **81.8** ±**0.1** |
| URM (DFR) | **89.7** ±**0.2** | **91.7** ±**0.1** | 67.4 ±0.8 | 93.5 ±0.1 | 61.3 ±1.0 | **81.5** ±**0.0** | **70.7** ±**0.9** | 80.8 ±0.1 |

### 3.5 Uniformly Distributed Feature Representations Improve Domain Generalization

URM improves the generalization performance of models in novel domains. URM outperformed competing methods in the RotatedMNIST and VLCS and PACS benchmarks (Table 3). Moreover, when combined with

Table 3: Domain generalization accuracies using leave-one-domain-out validation. Results were averaged over three random seeds. Results for other methods are from Gulrajani & Lopez-Paz (2021).

| Algorithm | ColoredMNIST | RotatedMNIST | VLCS | PACS | OfficeHome | TerraIncognita |
|---|---|---|---|---|---|---|
| ERM | $36.7 \pm 0.1$ | $97.7 \pm 0.0$ | $77.2 \pm 0.4$ | $83.0 \pm 0.7$ | $65.7 \pm 0.5$ | $41.4 \pm 1.4$ |
| IRM | $40.3 \pm 4.2$ | $97.0 \pm 0.2$ | $76.3 \pm 0.6$ | $81.5 \pm 0.8$ | $64.3 \pm 1.5$ | $41.2 \pm 3.6$ |
| GroupDRO | $36.8 \pm 0.1$ | $97.6 \pm 0.1$ | $77.9 \pm 0.5$ | $83.5 \pm 0.2$ | $65.2 \pm 0.2$ | $44.9 \pm 1.4$ |
| Mixup | $33.4 \pm 4.7$ | $97.8 \pm 0.0$ | $77.7 \pm 0.6$ | $83.2 \pm 0.4$ | $67.0 \pm 0.2$ | $48.7 \pm 0.4$ |
| MLDG | $36.7 \pm 0.2$ | $97.6 \pm 0.0$ | $77.2 \pm 0.9$ | $82.9 \pm 1.7$ | $66.1 \pm 0.5$ | $46.2 \pm 0.9$ |
| CORAL | $39.7 \pm 2.8$ | $97.8 \pm 0.1$ | $78.7 \pm 0.4$ | $82.6 \pm 0.5$ | $68.5 \pm 0.2$ | $46.3 \pm 1.7$ |
| MMD | $36.8 \pm 0.1$ | $97.8 \pm 0.1$ | $77.3 \pm 0.5$ | $83.2 \pm 0.2$ | $60.2 \pm 5.2$ | $46.5 \pm 1.5$ |
| DANN | $\mathbf{40.7 \pm 2.3}$ | $97.6 \pm 0.2$ | $76.9 \pm 0.4$ | $81.0 \pm 1.1$ | $64.9 \pm 1.2$ | $44.4 \pm 1.1$ |
| CDANN | $39.1 \pm 4.4$ | $97.5 \pm 0.2$ | $77.5 \pm 0.2$ | $78.8 \pm 2.2$ | $64.3 \pm 1.7$ | $39.9 \pm 3.2$ |
| MTL | $35.0 \pm 1.7$ | $97.8 \pm 0.1$ | $76.6 \pm 0.5$ | $83.7 \pm 0.4$ | $65.7 \pm 0.5$ | $44.9 \pm 1.2$ |
| SagNet | $36.5 \pm 0.1$ | $94.0 \pm 3.0$ | $77.5 \pm 0.3$ | $82.3 \pm 0.1$ | $67.6 \pm 0.3$ | $47.2 \pm 0.9$ |
| ARM | $36.8 \pm 0.0$ | $\mathbf{98.1 \pm 0.1}$ | $76.6 \pm 0.5$ | $81.7 \pm 0.2$ | $64.4 \pm 0.2$ | $42.6 \pm 2.7$ |
| VREx | $36.9 \pm 0.3$ | $93.6 \pm 3.4$ | $76.7 \pm 1.0$ | $81.3 \pm 0.9$ | $64.9 \pm 1.3$ | $37.3 \pm 3.0$ |
| RSC | $36.5 \pm 0.2$ | $97.6 \pm 0.1$ | $77.5 \pm 0.5$ | $82.6 \pm 0.7$ | $65.8 \pm 0.7$ | $40.0 \pm 0.8$ |
| URM | $36.9 \pm 0.2$ | $\mathbf{98.1 \pm 0.2}$ | $\mathbf{84.3 \pm 4.8}$ | $\mathbf{84.0 \pm 0.2}$ | $67.5 \pm 0.5$ | $48.3 \pm 1.4$ |
| URM (Mixup) | $32.4 \pm 3.6$ | $96.7 \pm 0.4$ | $77.1 \pm 0.2$ | $\mathbf{87.2 \pm 3.4}$ | $\mathbf{68.9 \pm 0.6}$ | $\mathbf{49.3 \pm 0.9}$ |

Inter-domain Mixup, URM achieved the best accuracy on the PACS dataset achieving 3.5% better accuracy than the best competing method, as well as the OfficeHome and Terraincognita benchmarks (Table 3). This demonstrates that URM, a robust representation learning method, can be combined with an orthogonal method such as Mixup, a data augmentation method. URM does not use any knowledge of the domain generalization task and yet achieves competitive accuracy with methods that do. As URM is performed by learning a feature representation that is uniformly distributed over a bounded range e.g. $[-1, 1]^Z$, where $Z$ is the dimensionality of the feature space, feature representations of test domain samples fall within the familiar feature space of the URM model, reducing the likelihood of unexpected outputs. Hence, the URM model sees the entire range of feature representations equally during training, improving domain generalization.

### 3.6 Visualizing Uniformity of Learned Feature Representations

Data distributions often have multiple peaks and troughs, and models trained on such distributions tend to be biased towards high-density regions (majority groups) since they contribute more to gradient updates during training. Consequently, the learned parameters are more optimized for these high-density regions, resulting in poorer performance on low-density regions (the troughs). URM addresses this by learning a feature space that is more uniformly distributed, effectively flattening the training feature distribution and reducing bias. This approach encourages the model to learn features that are more evenly distributed across both high- and low-density regions. To demonstrate this, we visualized the feature representations learned by URM compared to those learned by ERM using UMAP (McInnes et al., 2018) (Figure 2). We extracted feature vectors for test samples in the Waterbirds dataset using ERM and URM trained models. While UMAP does not provide a complete picture of the data due to significant dimensionality reduction, we observed that URM's features were significantly more uniformly distributed than ERM's. This suggests that URM achieves the training objective for the feature representations to be more uniformly distributed and contributes to the robustness of the model.

## 4 Related Work

**Group robustness:** The topic of robustness to sub-population shifts has been increasingly studied in recent years as machine learning models are being deployed in real-world applications where we wish to avoid downstream performance biases. Many such methods involve training on a group balanced dataset with improved performance. As shown above, URM in fact supports this approach of training on uniform

**Empirical Risk Minimization**    **Uniform Risk Minimization**

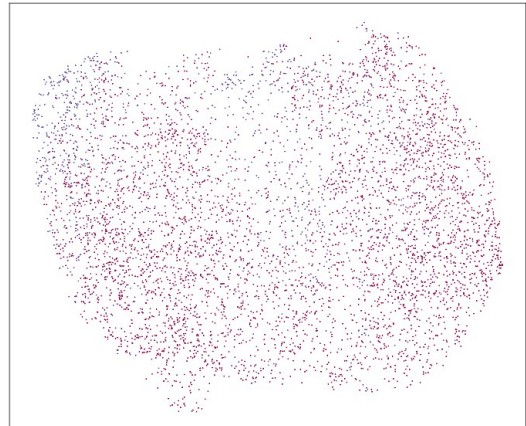

Figure 2: UMAP (McInnes et al., 2018) visualization of feature vectors generated by ERM and URM on Waterbirds dataset. URM generates a more uniformly distributed feature representation space compared to ERM.

group distributions. For example, one of the existing best methods, DFR, retrains the linear classifier head on a group balanced dataset. However, group-attributes may not available if many practical scenarios. Hence, a method such as URM that does not require group annotations would be useful. Moreover, as shown above, URM can also be combined with methods such as DFR for improved performance (Section 3.4).

**Uniform priors:** We note that our work is closely related to the ideas presented in uniform priors for data-efficient learning (Sinha et al., 2022). Compared to (Sinha et al., 2022), we proposed a novel learning principle and framework to motivate the use of uniform priors for robustness. In fact, our results may explain the empirical results of (Sinha et al., 2022). In other work, uniform priors have also been employed in self-supervised representation learning, both explicitly and implicitly (Assran et al., 2023).

**Robust Representation Learning:** URM is also related to the idea of state reification (Lamb et al., 2019), which aims to project out-of-distribution hidden states back on to the manifold of familiar hidden states from the training distribution. Rather than training additional networks to map out-of-distribution hidden states back on to the training data manifold, URM learns bounded feature spaces that are uniformly distributed. Hence, even out-of-distribution samples will fall in the familiar feature space of the model and is less likely to provide unexpected outputs. Other work has introduced input-space transformations that mitigate the effect of irrelevant input features to improve representation learning (Taghanaki et al., 2021).

## 5    Conclusion

We presented a novel learning principle called Uniform Risk Minimization (URM) and showed theoretically that uniform training data distributions are an optimal choice to lower *uniform risk*, a novel risk measure for ood robustness. Our results also provide a unified perspective on various existing algorithms that perform class-balanced or group-balanced training from the perspective of distribution shift. Finally, we proposed an empirical method to learn uniformly distributed feature representations in deep neural networks using adversarial distribution matching. We empirically demonstrated the effectiveness of our method to improve robustness to sub-population shifts as well as domain generalization, without knowledge of any privileged group or domain attribute information during training. Our work sheds light on the importance of the distribution of learned feature representations for model robustness and fairness. We hope our work encourages further research into uniform data distributions and priors for robust and fair representation learning.

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

# A  Proof of Proposition 2.1

*Proof.*

$$l_{test} \leq l_{train} + \frac{M}{\sqrt{2}}\sqrt{\text{KL}[p_{te}(x,y)|p_{tr}(x,y)]} \qquad \text{(using Proposition 1 of (Nguyen et al., 2022))}$$

$$\leq l_{train} + \frac{M}{\sqrt{2}}[\text{KL}[p_{te}(x,y)|p_{tr}(x,y)] + \frac{1}{4}] \qquad \text{(square the square root}^2\text{)}$$

$$= l_{train} + \frac{M}{\sqrt{2}}\text{KL}[p_{te}(x,y)|p_{tr}(x,y)] + \frac{M}{4\sqrt{2}} \tag{7}$$

$$= l_{train} + \frac{M}{\sqrt{2}}\mathbb{E}_{p_{te}(x,y)}[\log p_{te}(x) + \log p_{te}(y|x) - \log p_{tr}(x) - \log p_{tr}(y|x)] + \frac{M}{4\sqrt{2}} \tag{8}$$

$$= l_{train} + \frac{M}{\sqrt{2}}\mathbb{E}_{p_{te}(x,y)}[\log p_{te}(x) - \log p_{tr}(x)$$

$$+ \mathbb{E}_{p_{te}(x,y)}[\log p_{te}(y|x) - \log p_{tr}(y|x)]] + \frac{M}{4\sqrt{2}} \tag{9}$$

$$= l_{train} + \frac{M}{\sqrt{2}}\mathbb{E}_{p_{te}(x)}[\log p_{te}(x) - \log p_{tr}(x)]$$

$$+ \frac{M}{\sqrt{2}}\mathbb{E}_{p_{te}(x)}\left[\mathbb{E}_{p_{te}(y|x)}[\log p_{te}(y|x) - \log p_{tr}(y|x)]\right] + \frac{M}{4\sqrt{2}} \tag{10}$$

$$= l_{train} + \frac{M}{\sqrt{2}}[\text{KL}[p_{te}(x)|p_{tr}(x)] + \mathbb{E}_{p_{te}(x)}[\text{KL}[p_{te}(y|x)|p_{tr}(y|x)]]] + \frac{M}{4\sqrt{2}} \tag{11}$$

$$= l_{train} + \frac{M}{\sqrt{2}}[\text{KL}[p_{te}(x)|p_{tr}(x)] + \frac{M}{4\sqrt{2}} \qquad \{\text{covariate-shift}\} \tag{12}$$

$$\square$$

Using the *covariate-shift* assumption, the second KL divergence term in Eq. 11 involving the conditional probabilities $p(y|x)$ equals 0 and can be dropped. This is a reasonable assumption as generalization to distributions different from the training distribution is challenging if the labeling mechanism changes (Ben-David et al., 2010). Hence, we can make the common assumption that $\text{KL}[p_{te}(y|x)|p_{tr}(y|x)] = 0$. We can then focus on the divergence between the marginal distributions i.e. $\text{KL}[p_{te}(x)|p_{tr}(x)]$.

# B  Proof of Proposition of 2.2

*Proof.*

$$\underset{p_{tr}(x)}{\arg\min}\, \mathbb{E}_{Dir(\boldsymbol{\alpha}=\mathbb{1})}[\text{KL}[p_{te}(x)|p_{tr}(x)]] \tag{13}$$

$$= \underset{p_{tr}(x)}{\arg\min}\, \mathbb{E}_{Dir(\boldsymbol{\alpha}=\mathbb{1})}[\sum_{i=1}^{N} p_{te}(x_i)ln(\frac{p_{te}(x_i)}{p_{tr}(x_i)})] \tag{14}$$

$$= \underset{p_{tr}(x)}{\arg\min}\, \mathbb{E}_{Dir(\boldsymbol{\alpha}=\mathbb{1})}[\sum_{i=1}^{N} p_{te}(x_i)ln(p_{te}(x_i)) - p_{te}(x_i)ln(p_{tr}(x_i))] \tag{15}$$

$$= \underset{p_{tr}(x)}{\arg\min}\, \mathbb{E}_{Dir(\boldsymbol{\alpha}=\mathbb{1})}[\sum_{i=1}^{N} p_{te}(x_i)ln(p_{te}(x_i)) - \sum_{i=1}^{N} p_{te}(x_i)ln(p_{tr}(x_i))] \tag{16}$$

$$= \underset{p_{tr}(x)}{\arg\min}\, \mathbb{E}_{Dir(\boldsymbol{\alpha}=\mathbb{1})}[\sum_{i=1}^{N} p_{te}(x_i)ln(p_{te}(x_i))] - \mathbb{E}_{Dir(\boldsymbol{\alpha}=\mathbb{1})}[\sum_{i=1}^{N} p_{te}(x_i)ln(p_{tr}(x_i))] \tag{17}$$

---

[2]Note that KL-divergence is non-negative. $\frac{1}{4}$ is added as a buffer in case the KL-divergence is less than 1, as $\forall x >= 0, x + \frac{1}{4} \geq \sqrt{x}$.

$$= \underset{p_{tr}(x)}{\arg\min} \sum_{i=1}^{N} \mathbb{E}_{Dir(\boldsymbol{\alpha}=\mathbb{1})}[p_{te}(x_i)ln(p_{te}(x_i))] - \sum_{i=1}^{N} \mathbb{E}_{Dir(\boldsymbol{\alpha}=\mathbb{1})}[p_{te}(x_i)ln(p_{tr}(x_i))] \tag{18}$$

$$= \underset{p_{tr}(x)}{\arg\min} - \sum_{i=1}^{N} \mathbb{E}_{Dir(\boldsymbol{\alpha}=\mathbb{1})}[p_{te}(x_i)ln(p_{tr}(x_i))] \qquad \text{(remove terms independent of } p_{tr}(x_i))$$

$$= \underset{p_{tr}(x)}{\arg\min} - \sum_{i=1}^{N} ln(p_{tr}(x_i))\mathbb{E}_{Dir(\boldsymbol{\alpha}=\mathbb{1})}[p_{te}(x_i)] \qquad \text{(expectation of Dirichlet dist. is } \tfrac{\alpha_i}{\alpha_0})$$

$$= \underset{p_{tr}(x)}{\arg\min} - \sum_{i=1}^{N} ln(p_{tr}(x_i))[\frac{\alpha_i}{\alpha_0}] \qquad (\alpha_i = 1,\ \alpha_0 = N)$$

$$= \underset{p_{tr}(x)}{\arg\min} - \frac{1}{N} \sum_{i=1}^{N} ln(p_{tr}(x_i)) \tag{19}$$

$$\therefore \underset{p_{tr}(x)}{\arg\min} \mathbb{E}_{Dir(\boldsymbol{\alpha}=\mathbb{1})}[\text{KL}[p_{te}(x)|p_{tr}(x)]] = \underset{p_{tr}(x)}{\arg\min} - \frac{1}{N} \sum_{i=1}^{N} ln(p_{tr}(x_i)) \tag{20}$$

$$\text{subject to } \sum_{i=1}^{N} p_{tr}(x_i) = 1$$

The Lagrangian for this constrained optimization problem is:

$$\mathcal{L}[p_{tr}(x_1), \ldots, p_{tr}(x_N), \lambda] = -\frac{1}{N} \sum_{i=1}^{N} ln(p_{tr}(x_i)) + \lambda(\sum_{i=1}^{N} p_{tr}(x_i) - 1) \tag{21}$$

$$\text{solve } \nabla_{p_{tr}(x_1), \ldots, p_{tr}(x_N), \lambda} \mathcal{L}[p_{tr}(x_1), \ldots, p_{tr}(x_N), \lambda] = 0$$

$$\Rightarrow p_{tr}(x_i) = \frac{1}{N\lambda}, \forall i \in \{1, \ldots, N\}$$

$$\Rightarrow \lambda = 1, \because \sum_{i=1}^{N} p_{tr}(x_i) = 1 \text{ (constraint)} \tag{22}$$

$$\Rightarrow p_{tr}(x_i) = \frac{1}{N}, \forall i \in \{1, \ldots, N\} \text{ (i.e. uniform distribution)} \tag{23}$$

$$\therefore \underset{p_{tr}(x)}{\arg\min} \mathbb{E}_{p_{te} \sim Dir(\boldsymbol{\alpha}=\mathbb{1})}[\text{KL}[p_{te}(x)|p_{tr}(x)]] = u_{tr}^*(x) \tag{24}$$

$$\text{where } u_{tr}^*(x_i) = \frac{1}{N}, \forall i \in \{1, \ldots, N\}.$$

$\square$

## C   Proof of Proposition of E.1

*Proof.*

$$l_{test} \leq l_{train} + \frac{M}{\sqrt{2}} \sqrt{\text{KL}[p_{te}(x,y)|p_{tr}(x,y)]} \qquad \text{(using Proposition 1 of (Nguyen et al., 2022))}$$

$$\leq l_{train} + \frac{M}{\sqrt{2}}[\text{KL}[p_{te}(x,y)|p_{tr}(x,y)] + \frac{1}{4}] \qquad \text{(square the square root}^3)$$

$$= l_{train} + \frac{M}{\sqrt{2}}\text{KL}[p_{te}(x,y)|p_{tr}(x,y)] + \frac{M}{4\sqrt{2}} \tag{25}$$

$$= l_{train} + \frac{M}{\sqrt{2}}\mathbb{E}_{p_{te}(x,y)}[\log p_{te}(y) + \log p_{te}(x|y) - \log p_{tr}(y) - \log p_{tr}(x|y)] + \frac{M}{4\sqrt{2}} \tag{26}$$

$$
\begin{aligned}
&= l_{train} + \frac{M}{\sqrt{2}} \mathbb{E}_{p_{te}(x,y)}[\log p_{te}(y) - \log p_{tr}(y) \\
&\quad + \mathbb{E}_{p_{te}(x,y)}[\log p_{te}(x|y) - \log p_{tr}(x|y)]] + \frac{M}{4\sqrt{2}}
\end{aligned} \tag{27}
$$

$$
\begin{aligned}
&= l_{train} + \frac{M}{\sqrt{2}} \mathbb{E}_{p_{te}(y)}[\log p_{te}(y) - \log p_{tr}(y)] \\
&\quad + \frac{M}{\sqrt{2}} \mathbb{E}_{p_{te}(y)} \left[ \mathbb{E}_{p_{te}(x|y)}[\log p_{te}(x|y) - \log p_{tr}(x|y)] \right] + \frac{M}{4\sqrt{2}}
\end{aligned} \tag{28}
$$

$$
= l_{train} + \frac{M}{\sqrt{2}} [\mathrm{KL}[p_{te}(y)|p_{tr}(y)] + \mathbb{E}_{p_{te}(y)} \left[ \mathrm{KL}[p_{te}(x|y)|p_{tr}(x|y)]] \right] + \frac{M}{4\sqrt{2}} \tag{29}
$$

$$
= l_{train} + \frac{M}{\sqrt{2}} [\mathrm{KL}[p_{te}(y)|p_{tr}(y)] + \frac{M}{4\sqrt{2}} \qquad \{\text{target-shift}\} \tag{30}
$$

$\square$

In Eq. 29, we assume that the conditional distribution of $p(x|y)$ does not change and that only the label distribution $p(y)$ has shifted. This assumes that the generative process $p(x|y)$ has not changed at test time. This is referred to *target shift* in the domain adaptation literature (Zhang et al., 2013). Hence, we can drop the second term involving the conditional misalignment is zero and focus on the target shift in Eq. 30. Based on Eq. 30, we can similarly show that the expected test loss under target shift is minimized by the uniform training class-distribution.

## D  Extending Propositions 2.1 and 2.2 to Deep Feature Layers

Here we extend Propositions 2.1 and 2.2 to feature spaces such as those learned by a deep neural network. This will be useful below when we wish to apply the above results to a method that modifies the feature space to be uniformly distributed in order to improve robustness. Let $G$ be a model parameterized by a deep neural network that outputs a latent variable $z$ i.e. $z = G(x)$, which can then be used for a downstream task such as classification. To extend the propositions to the latent space, $z$, we make additional assumptions about the input data $x, y$ and the latent variable, $z$:

**Assumption D.1.** $I_{tr}(z, y) = I_{tr}(x, y)$, where $I(\cdot, \cdot)$ is mutual information, defined as:

$$
I_{tr}(z, y) = \mathbb{E}_{p_{tr}(z,y)} \left[ \log \frac{p_{tr}(z,y)}{p_{tr}(z)p_{tr}(y)} \right]; \quad I_{tr}(x, y) = \mathbb{E}_{p_{tr}(x,y)} \left[ \log \frac{p_{tr}(x,y)}{p_{tr}(x)p_{tr}(y)} \right] \tag{31}
$$

This is a reasonable assumption that the latent variable $z$ is *sufficient* to solve the task and contains the same information about the output variable $y$ as does the input variable $x$. As the latent representation $z$ is learned while optimizing the task objective, it must learn to retain information about the label.

**Assumption D.2.** $p_{tr}(y|x) = \mathbb{E}_{p(z|x)}[p_{tr}(y|z)], \quad \forall x, y \in \mathcal{X}, \mathcal{Y}$

This assumption simply allows us to re-write the true predictive distribution $p_{tr}(y|x)$ as a conditional expectation over the latent variable $z$. When this assumption holds, the predictive distribution, $\hat{p}(y|x) = \mathbb{E}_{p(z|x)}[\hat{p}(y|z)]$, will approximate $p_{tr}(y|x)$ when $\hat{p}(y|z) = p_{tr}(y|z)$, which is a goal of training.

**Assumption D.3.** $\frac{p_{te}(x,y)}{p_{tr}(x,y)} < \infty, \quad \forall x, y \in \mathcal{X}, \mathcal{Y}$

This assumption is satisfied when train and test distributions have the same support set $\mathcal{X}, \mathcal{Y}$. This is a reasonable assumption for sub-population shift scenarios when the test distribution is assumed to comprise the same groups as the training distribution but in different proportions.

**Assumption D.4.** $\mathrm{KL}[p_{te}(y|x)|p_{tr}(y|x)] = 0$

This is the covariate-shift assumption.

---

[3]Note that KL-divergence is non-negative. $\frac{1}{4}$ is added as a buffer in case the KL-divergence is less than 1, as $\forall x >= 0, x + \frac{1}{4} \geq \sqrt{x}$.

Using these four assumptions, Proposition 1 of (Nguyen et al., 2022) can be re-written as follows (see (Nguyen et al., 2022) for proofs):

$$l_{test} \le l_{train} + \frac{M}{\sqrt{2}}[\text{KL}[p_{te}(z)|p_{tr}(z)] + \frac{M}{4\sqrt{2}} \tag{32}$$

We have thus replaced $x$ (input space) with $z$ (latent space). The rest of our claims in Propositions 2.1 and 2.2 can then straightforwardly applied to the latent space rather than the input space.

## E   Uniform Risk for Label Shifts and Sub-population Shifts

In the context of sub-population and label shifts, *uniform risk* can be shown to be the balanced risk across groups or classes, respectively.

Let $\mathcal{R}_g$ be the group-conditional risk computed over each group-specific distribution $p_g \coloneqq p(x, y|g)$ i.e. $\mathcal{R}_g \coloneqq \mathbb{E}_{p_g}[l(x, y)]$. In the case of *label-shift*, the group can be treated as a class instead. Assume that the distribution shift is strictly over the group or class distribution.

$$
\begin{aligned}
\mathcal{R}_U &\coloneqq \mathbb{E}_{p_{te} \sim Dir(\boldsymbol{\alpha}=\mathbb{1})}[l_{\text{test}}] \\
&= \mathbb{E}_{p_{te} \sim Dir(\boldsymbol{\alpha}=\mathbb{1})}[\mathbb{E}_{p_g}[\mathcal{R}_g]] \\
&= \mathbb{E}_{p_{te} \sim Dir(\boldsymbol{\alpha}=\mathbb{1})}[\sum_{g \in \mathcal{G}} \beta_g \mathcal{R}_g], \text{where } \beta \in \Delta_{|\mathcal{G}|} \\
&= \sum_{g \in \mathcal{G}} \mathcal{R}_g \mathbb{E}_{p_{te} \sim Dir(\boldsymbol{\alpha}=\mathbb{1})}[\beta_g] \\
&= \frac{1}{|\mathcal{G}|} \sum_{g \in \mathcal{G}} \mathcal{R}_g \qquad\qquad (balanced\ risk)
\end{aligned}
$$

Thus, *uniform risk* is the balanced risk of a model over groups or classes (Eq. *balanced risk*) in the context of sub-population or label shifts. However, in the sub-population shift literature, authors generally focus on the worst-group performance. The limitations of over-emphasis on the *worst-group accuracy* metric has been reported in prior work (Section 5.5 of Yang et al. (2023)). In contrast, *uniform risk* provides a balanced risk measure that equally weighs all groups or classes.

### E.1   Class Balancing Lowers Uniform Risk

Here we discuss the common practice of class-balancing datasets from the perspective of *uniform risk* and robustness to *label* shifts at test time. In Eq. 7 of the Proof of Proposition 2.1, we can alternatively decompose the KL-divergence between the joint training and test distributions $p(x, y)$ using the marginal misalignment in $p(y)$ and the conditional misalignment in $p(x|y)$. We can then similarly show that training on data with a uniform *class or label* distribution minimizes the upper bound on *uniform risk*, as pertaining to the *label-shift* problem.

**Proposition E.1.** *If the loss* $-\log \hat{p}(y|x)$ *is bounded by* $M$ *and the generative process,* $p(x|y)$, *is the same between train and test distributions[4], we have:*

$$l_{test} \le l_{train} + \frac{M}{\sqrt{2}}\text{KL}[p_{te}(y)|p_{tr}(y)] + \frac{M}{4\sqrt{2}} \tag{33}$$

*Proof.* Proof in Appendix C.                                                                            □

---

[4]This is known as the *label* or *target shift* assumption in the distribution-shift literature.

The expectation of Eq. 33 with respect to a uniform prior distribution over test class-distributions is:

$$\mathbb{E}_{p_{te} \sim Dir(\boldsymbol{\alpha}=\mathbb{1})}[l_{test}] \leq l_{train} + \frac{M}{\sqrt{2}} \mathbb{E}_{p_{te}(y) \sim Dir(\boldsymbol{\alpha}=\mathbb{1})}[\text{KL}[p_{te}(y)|p_{tr}(y)]] + \frac{M}{4\sqrt{2}} \tag{34}$$

We can now aim to minimize the upper bound on the expectation of test-loss under label-shift i.e. *uniform risk* for label-shifts, by minimizing the expected KL-divergence between train and test class distributions. Let $C$ be the number of classes in the classification task. The proof follows the same steps as the proof of Proposition 2.2 but for the class-distribution p(y) instead of p(x).

**Proposition E.2.** *The expected KL-divergence between train and test class distributions in Eq. 4 is minimized when the training class distribution $p_{tr}(y)$ is uniform:*

$$\underset{p_{tr}(y)}{\arg\min} \, \mathbb{E}_{p_{te}(y) \sim Dir(\boldsymbol{\alpha}=\mathbb{1})}[\text{KL}[p_{te}(y)|p_{tr}(y)]] = u^*_{tr}(y) \tag{35}$$

$$where \; u^*_{tr}(y_i) = \frac{1}{C}, \forall i \in \{1, \ldots, C\}.$$

*Proof.* Same as proof of Proposition 2.2. $\qquad\square$

Therefore our results support the common practice of class-balancing from the perspective of robustness under label shifts. Indeed, prior work has demonstrated the efficacy of balancing classes during training (Buda et al., 2018). We further experimentally demonstrate this below (Section 3.3).

## F  Datasets

We describe each dataset used in our group robustness benchmarks below.

- **Waterbirds (Wah et al., 2011).** Waterbirds is a binary classification image dataset containing spurious correlations, constructed by placing images from the Caltech-UCSD Birds-200-2011 (CUB) dataset (Wah et al., 2011) over backgrounds from the Places dataset (Zhou et al., 2018). The task is to classify landbirds from waterbirds and the spurious attribute is the background (water or land). As most images of waterbirds have a water background and most images of landbirds have a land background, models may latch on to the spurious correlation between the background and the type of bird.

- **CelebA (Liu et al., 2015).** CelebA is a binary classification image dataset. The task is to predict hair color from images of celebrity faces (blond *vs.* non-blond), where the spurious correlation is gender. As most blond haired people in this dataset are female and most non-blond haired people are male, this creates a spurious correlation between gender and hair color.

- **CivilComments (Borkan et al., 2019).** CivilComments is a binary classification text dataset where models must predict whether an internet comment contains toxic language. The spurious attribute is the presence of references to eight demographic identities (*male, female, LGBTQ, Christian, Muslim, other religions, Black,* and *White*).

- **MultiNLI (Williams et al., 2018).** MultiNLI is a text classification dataset with 3 classes and the target is the natural language inference relationship between the premise and the hypothesis (neutral, contradiction, or entailment). The spurious attribute is the presence of negation in the text, as negation is highly correlated with the contradiction label.

**Domain Generalization Datasets:**  We benchmarked on multiple challenging DG datasets. ColoredMNIST (Arjovsky et al., 2020) is a variant of the MNIST digit recognition dataset where each domain contains a disjoint set of digits colored either red or blue. The binary label is a noisy function of digit and color, such that color is correlated with the label to varying degree in each domain and the digit bears correlation 0.75 with the label. RotatedMNIST (Ghifary et al., 2015) is also a variant of MNIST where each domain contains

digits rotated to different degrees (0, 15, 30, 45, 60, 75 and contains 10 classes. PACS (Li et al., 2017) contains four diverse domains (art, cartoons, photos, sketches) and 7 classes. VLCS (Fang et al., 2013) contains four photographic domains (Caltech101, LabelMe, SUN09, VOC2007) and 5 classes. OfficeHome (Venkateswara et al., 2017) also includes four domains (art, clipart, product, real) and 65 classes. TerraIncognita (Beery et al., 2018) comprises photographs of animals taken by camera traps at four different locations.

## G   Baselines

**Vanilla Training:**   Empirical Risk Minimization (ERM, (Vapnik, 1998)), minimizes the errors across all training samples. **Subgroup Robust Methods:** Group distributionally robust optimization (GroupDRO) (Sagawa et al., 2020) performs ERM while emphasising sub-populations or domains with high losses during training. CVaRDRO (John C. Duchi & Hongseok Namkoong, 2021) is a variant of GroupDRO that up-weights training samples with the highest losses. LfF (Nam et al., 2020) trains two models where the first model is biased and the second one is debiased using a re-weighted objective. Just train twice (JTT) (Liu et al., 2021) first trains a standard ERM model to identify minority sub-populations in the dataset and then trains another ERM model while up-weighting minority samples. LISA (Yao et al., 2022) trains invariant predictors using data interpolation within and across attributes. Deep feature re-weighting (DFR) (Izmailov et al., 2022) first trains an ERM model and then retrains only the last layer of the model using a dataset balanced among different sub-populations. **Data Augmentation Method:** Mixup (Zhang et al., 2018) trains on linear interpolations of randomly sampled training data points and their labels. **Domain-Invariant Representation Learning Methods:** Invariant risk minimization (IRM) (Arjovsky et al., 2020) learns a feature space such that the optimal linear classifier is the same across domains. Deep correlation alignment (CORAL) (Sun & Saenko, 2016) matches the second order moments of the feature distributions. Maximum mean discrepancy (MMD) (Li et al., 2018b) matches the MMD (Gretton et al., 2012) of feature distributions across domains. Domain Adversarial Neural Networks (DANN, (Ganin et al., 2016)) employs adversarial training to match feature distributions across domains. Class-conditional DANN (C-DANN, (Li et al., 2018b)) is similar to DANN as it matches the class-conditional feature distribution across domains. **Imbalanced Learning Methods:** ReSample (Japkowicz, 2000) and ReWeight (Japkowicz, 2000) simply re-sample or re-weight the inputs according to the number of samples per class. Focal loss (Focal) (Lin et al., 2020) reduces the relative loss for well-classified samples and focuses on difficult samples. Class-balanced loss (CBLoss) (Cui et al., 2019) proposes re-weighting by the inverse effective number of samples. The LDAM loss (LDAM) (Cao et al., 2019) employs a modified marginal loss that favors minority samples more. Balanced-Softmax (BSoftmax) (Ren et al., 2020) extends Softmax to an unbiased estimation that considers the number of samples in each class. Classifier re-training (CRT) (Kang et al., 2020) decomposes the representation and classifier learning into two stages, where it fine-tunes the classifier using class-balanced sampling with representation fixed in the second stage. ReWeightCRT (Kang et al., 2020) is a re-weighting variant of CRT.

**Domain Generalization Baselines:**   Inter-domain Mixup (Mixup, (Zhang et al., 2018; Wang et al., 2020)) trains on linear iterpolations between samples from different domains. Meta-Learning for Domain Generalization (MLDG, (Li et al., 2018a)) uses MAML to meta-learn generalizing across domains. Marginal Transfer Learning (MTL, (Blanchard et al., 2011; 2021)) estimates a mean embedding for each domain that is passed as a second argument to the model. Adaptive Risk Minimization (ARM, (Zhang et al., 2021)) extends MTL using a separate embedding model. Style-Agnostic Networks (SagNet, (Nam et al., 2021)) trains models by keeping image content and randomizing style. Representation Self Challenging (RSC, (Huang et al., 2020)) learns robust models by iteratively pruning the most activated features. Risk Extrapolation (VREx, (Krueger et al., 2021)) approximates IRM using a variance penalty.

## H   Training Hyper-parameters

Table 4: Search ranges for URM-specific hyper-parameters. Other training parameters ranges are based on (Yang et al., 2023).

| Method | Parameter | Random Distribution |
|--------|-----------|---------------------|
| URM | lambda $\lambda$ | $\text{Uniform}(0, 0.2)$ |
| | generator output | $\text{RandomChoice}([TanH, ReLU])$ |
| | number of layers in discriminator | $\text{RandomChoice}([1, 2, 3])$ |
| | learning rate (for image or tabular datasets) | $10^{\text{Uniform}(-5,-3)}$ |
| | optimizer (for image or tabular datasets) | $\text{RandomChoice}([SGD])$ |
| | learning rate (text datasets) | $10^{\text{Uniform}(-6,-5)}$ |
| | optimizer (text datasets) | $\text{RandomChoice}([AdamW])$ |

