# OpenReview forum: "Uniformly Distributed Feature Representations for Fair and Robust Learning"
_TMLR — Accepted by TMLR_

### Review · Reviewer_VVDg · 2024-09-04

**Summary Of Contributions:**

The paper proposes a new learning principle called Uniform Risk Minimization as an alternative to Empirical Risk Minimization to ensure better fairness and robustness. The context is a classification task that can be subject to distribution shifts (covariate shifts, target shifts). In a first part, the paper provides a theoretical study on the Uniform Risk (UR) where the main result is proving that under some common assumptions, a uniform distribution in the training data minimizes the UR. This result is also proven for a uniform distribution for the learned feature representations of a neural network. Results are then corroborated by multiple experiments on various datasets, where a performance comparison with some known baseline models as well as some insights are provided.
Overall, the URM concept is novel, and proofs and results look promising.

**Audience:**

Yes

**Claims And Evidence:**

Yes

**Requested Changes:**

See weaknesses section for more details. Some additional points:
- Add the code for reproducibility
- Section 2.6: shouldn't it be $\lambda [E_{p_{tr}}(log(1 - D(G(x))) + E_{p_u}(log D(\bar{z}))) ] $ (i.e. missing parentheses after $\lambda$)? Also given that it is explained "The same $\lambda$ is applied to both update the discriminator and encoder so that one does not overpower the other"

**Strengths And Weaknesses:**

Strengths:
- URM concept is interesting and results look promising
- Paper is rigorous and proofs are clear and easy to follow
- Theoretical study provides a strong mathematical grounding for URM
- Training procedure (section 2.6) is elegant and offers a good pipeline to reuse for similar tasks

Weaknesses:
- Clarity: some assumptions / reasonings seem unclear, e.g. why choose the Dirichlet distribution in section 2.3
- Lack of qualitative insights: some insights are provided (Figure 2), but I would have wanted additional comments and interpretations on how URM compares to ERM in different settings. Likewise, the paper did not spend much time analyzing specifically robustness and fairness (especially the former). A good example would have been analyzing a classification task with adversarial noise, either from a theoretical or empirical perspective

---

> ### Author Response · Authors · 2024-10-03
> **Response**
>
> We thank the reviewer for their supportive comments and appreciating that our proposed approach is "rigorous" and "elegant". We address the concerns raised below:
>
> **Clarity: some assumptions / reasonings seem unclear, e.g. why choose the Dirichlet distribution in section 2.3**
>
> The Dirichlet distribution is a natural prior distribution for N dimensional distributions. We use the Dirichlet distribution to define an uninformative prior to reflect the uncertainty over the test distribution as we assume we do not have knowledge of the target test distribution, as this is common in real-world scenarios.
>
> **Lack of qualitative insights: some insights are provided (Figure 2), but I would have wanted additional comments and interpretations on how URM compares to ERM in different settings. Likewise, the paper did not spend much time analyzing specifically robustness and fairness (especially the former). A good example would have been analyzing a classification task with adversarial noise, either from a theoretical or empirical perspective**
>
> Thank you for the suggestion. We've added some interpretations on why URM demonstrates greater robustness than ERM in Section 3.6. In summary, data distributions often have multiple peaks and troughs, and models trained on such distributions tend to be biased towards high-density regions (majority groups) since they contribute more to gradient updates during training. Consequently, the learned parameters are more optimized for these high-density regions, resulting in poorer performance on low-density regions (the troughs). URM addresses this by learning a feature space that is more uniformly distributed, effectively flattening the training feature distribution and reducing bias. This approach encourages the model to learn features that are more evenly distributed across both high- and low-density regions, as shown in Figure 2.
>
> Thank you for suggesting a study of adversarial robustness, however it is outside the scope and claims of this paper so we will consider studying this direction in follow up work.
>
> **Add the code for reproducibility**
>
> We will post the code to github upon publication.
>
> **Section 2.6: shouldn't it be $\lambda [E_{p_{tr}}(log(1 - D(G(x))) + E_{p_u}(log D(\bar{z}))) ]$ (i.e. missing parentheses after )? Also given that it is explained "The same is applied to both update the discriminator and encoder so that one does not overpower the other"**
>
> Thank you for your question. As mentioned in the paper, the hyper-parameter $\lambda$ determines the weight for training the encoder and discriminator **to classify the feature outputs $z=G(x)$**. The loss term, $E_{p_{tr}}(log(1 - D(G(x)))$, is computed over the data distribution $p_{tr}(x)$ and applies to both encoder and discriminator as computing it requires the output of the encoder, $G(x)$, and the output of the discriminator, $D(G(x))$. Hence, $\lambda$ indeed applies to both encoder and discriminator as you have highlighted. The last loss term $ E_{p_u}(log D(\bar{z})))$ only applies to the discriminator when classifying uniform noise vectors randomly sampled from $p_u$. We do not apply $\lambda$ to this loss term. Hence, the equation in the paper is correct as it is.

---

### Review · Reviewer_ysik · 2024-09-16

**Summary Of Contributions:**

The authors introduce a learning principle called Uniform Risk Minimization (URM), which addresses robustness to distribution shifts in machine learning, especially when test distributions differ from training distributions. URM trains DNNs to produce uniformly distributed feature representations in the final activation layer using adversarial training to address sub-population shifts and domain generalization without requiring prior knowledge of groups or domains during training. Theoretical analysis and experimental results over different datasets are performed to validate the effectiveness of the proposed URM learning strategy.

**Audience:**

Yes

**Claims And Evidence:**

Yes

**Requested Changes:**

- Please try to discuss the weaknesses.
- Although there are many different baselines, the main characteristics of these methods are not clearly presented in Tables 2 and 3. It would be helpful for the authors to provide concise information about each baseline and reorganize the tables for better clarity.
- Minor issue: try to fix the layout of Figure 2 for page 5 and page 6

**Strengths And Weaknesses:**

Strengths:
- The introduction of Uniform Risk Minimization (URM) provides a somewhat novel perspective on handling distribution shifts.
- The authors offer rigorous theoretical support for their method, demonstrating that uniformly distributed training data or feature representations minimize uniform risk.
- The submission includes thorough experiments across multiple benchmarks (e.g., sub-population shifts, domain generalization), demonstrating the effectiveness of URM and its competitive performance with previous methods.

Weaknesses:
- The use of adversarial training to enforce uniformity in feature representations may introduce additional computational complexity and training instability.
- It appears that the improvement achieved by using URM alone over other methods is not highly significant. Additionally, the comparisons in Table 2 (URM + DFR vs. others) and Table 3 (URM + Mixup vs. others) may not be entirely fair, as they compare combination methods against standalone approaches. If the authors choose to include combination methods, they should consider incorporating other combinations of the baseline methods for a more balanced comparison.

---

> ### Author Response · Authors · 2024-10-02
> **Response**
>
> We thank the reviewer for their supportive comments and for appreciating the novelty, theoretical rigor and empirical effectiveness of our approach. We address the concerns raised below:
>
> **The use of adversarial training to enforce uniformity in feature representations may introduce additional computational complexity and training instability.**
>
> Thanks for raising this question. Adversarial training is just one method to match distributions and we are aware it is not known to be always stable, but we did not experience significant instability in our experiments as we are not trying to generate high-dimensional images as in GANs (Generative Adversarial Networks), which is a more challenging distribution matching task. We are generating lower-dimensional feature vectors to match a uniform distribution. Adversarial training also provides some advantages as it requires minimal changes to the encoder architecture. Adversarial Autoencoders [ICLR 2015] have previously used this approach to match the latent distribution in VAEs (variational autoencoders) to an arbitrary prior distribution. If adversarial training is still not preferred, other methods to match the distribution can be explored.
>
> **It appears that the improvement achieved by using URM alone over other methods is not highly significant. Additionally, the comparisons in Table 2 (URM + DFR vs. others) and Table 3 (URM + Mixup vs. others) may not be entirely fair, as they compare combination methods against standalone approaches. If the authors choose to include combination methods, they should consider incorporating other combinations of the baseline methods for a more balanced comparison.**
>
> Thanks for raising this question. In domain generalization experiments, standalone URM is indeed competitive with existing methods even when it is not combined with MixUp (it outperforms or at par with the best existing method on 4/5 datasets). We combined it with Mixup only to demonstrate that URM (a representation learning method) can be combined with an orthogonal method such as Mixup (a data augmentation method).
>
> In sub-group robustness experiments, we combined URM with DFR because DFR involves fine-tuning the classifier on a group balanced validation set, which URM does not require as it is a generic representation learning method. Excluding methods that require fine-tuning the model on a group balanced validation set, standalone URM is indeed competitive with other methods.
>
> **Although there are many different baselines, the main characteristics of these methods are not clearly presented in Tables 2 and 3. It would be helpful for the authors to provide concise information about each baseline and reorganize the tables for better clarity.**
>
> Thank you for the suggestion but as the baselines are quite diverse and do not all fit into a category, we were not able to include this information in the tables. However, we have described each baseline in Appendix G and have tried to group the methods as much as possible e.g. "Data Augmentation Methods" and "Invariant Representation Learning Methods".
>
> **Minor issue: try to fix the layout of Figure 2 for page 5 and page 6**
>
> Thank you for pointing this out, we have fixed the layout of the figure.

---

### Review · Reviewer_2jfY · 2024-09-23

**Summary Of Contributions:**

The paper first proves that training on uniformly distributed training samples mitigates the distribution shifts problem, then generalizes the result on samples to features to show that the features should also be uniformly distributed. To learn a uniform feature distribution, adversarial learning is used and the experiment shows that the proposed URM method is effective in several datasets with distribution shifts.

**Audience:**

Yes

**Claims And Evidence:**

No

**Requested Changes:**

Please address the weaknesses in the above section.

**Strengths And Weaknesses:**

Strengths:

The empirical result shows the strength of the approach.

The proposed algorithm is presented clearly.


Weaknesses:

The assumption I(z,y)=I(x,y) is somehow problematic. This assumption indicates that the latent variable is sufficient to solve the “task”, which is a fixed distribution in the training environment. Then in the proof, the training distribution changes somehow, which leads to a circular argument.

Though the theoretical result is intuitively correct, it is probably not quite informative as it only shows how to minimize the KL divergence between the training and test distribution without considering the utility of the model. In other words, the fair representation will be achieved if the model outputs uniformly distributed noise for any input, but that representation is not useful at all.

---

> ### Author Response · Authors · 2024-10-01
> **Response**
>
> We thank the reviewer for their supportive comments and acknowledging the strength of our proposed method. We address the concerns raised below:
>
> **The assumption I(z,y)=I(x,y) is somehow problematic. This assumption indicates that the latent variable is sufficient to solve the “task”, which is a fixed distribution in the training environment. Then in the proof, the training distribution changes somehow, which leads to a circular argument.**
>
> Thanks for raising this question. We would like to clarify that we do not change the training environment, we instead find the optimal latent variable distribution $p_{train}(z)$. The latent variable $z$ is not fixed or unique and is learned by a model for a given training distribution. Hence, the latent variable distribution $p_{train}(z)$ can be modified using a regularizer during training.
>
> **Though the theoretical result is intuitively correct, it is probably not quite informative as it only shows how to minimize the KL divergence between the training and test distribution without considering the utility of the model. In other words, the fair representation will be achieved if the model outputs uniformly distributed noise for any input, but that representation is not useful at all**
>
> Thanks for your question. The assumption I(x,y) = I(z,y) ensures that the representation is still useful for classification. Moreover, we would like to clarify that a variable that has a uniform distribution need not be just random noise. In practice, this is ensured by the joint-optimization of the classifier and the regularizer. Hence, $z$ is learned to be a useful representation in addition to being uniformly distributed. A relevant and similar approach from the literature are Adversarial Autoencoders [ICLR 2015] where the latent variable is jointly optimized to reconstruct the input and to match an arbitrary prior distribution. In Adversarial Autoencoders, this learns a representation that is useful for reconstruction while matching the prior distribution.

---

### Decision · Action_Editor_VUxH · 2024-11-19

**Recommendation:** Accept with minor revision

**Comment:**

please see above for required changes in terms of clarification in the writeup in the theory section.

please address all the points that were not clear to the reviewers.

as promised, please upload the code to gitub for reproducibility.

Another point that would be nice to address is in terms of discussions on stability of the method when running experiments.

**Audience:**

ML community interested in distribution shifts, sub-population performance, generalization performance, OOD performance

**Claims And Evidence:**

The paper proposes URM(uniform risk minimization) as a framework to alleviate the issue of generalization for under-representated sub-populations. the ideas are first derived theoretically by showing that uniform training data distributions and feature representations support robustness to unknown distribution shifts.
Then motivated by these results an empirical method is suggested that learns a uniform representation in the final activation layer of a DNN for better robustness. the method is accompanied by experimental results showing its performance in practice.

The authors offer rigorous theoretical support for their method, demonstrating that uniformly distributed training data or feature representations minimize uniform risk. the theory is proven as well, but it needs to be clarified and explained better as also was raised by one of the reviewers.  the framework needs to be clearly explained and mentioned which part is considered uniform, the comparison to adversarial autoencoders will also help the reader. that said, these are for clarification and the claims are correct.